# FAIRNESS UNDER PARTIAL OBSERVABILITY

**Peter Hill**[1], **Islam Utyagulov**[1], and **Francois Buet-Golfouse**[1,2]

[1]Decision Science, JPMorgan Chase, London, UK
[2]University College London

## ABSTRACT

The purpose of this article is to discuss an important challenge faced in "real life" when trying to implement *group* fairness-aware models and algorithms. Here, we focus specifically on the role that uncertainty and ambiguity play and revisit the case where protected attributes are only partially observable.

## 1 PARTIAL OBSERVABILITY OF PROTECTED ATTRIBUTES

Protected attributes are rarely fully observable, so let us consider a situation where the protected attribute $S$ is not directly observable and can only be inferred via an observation variable, $Z$, which is a proxy for $S$. This is a setup that is very often encountered in practice in financial services (refer to (Bureau, 2014; Chen et al., 2019) for concrete examples and a discussion).

**Notations** In this article, we define $S \in \mathcal{S} = \{0, 1\}$ to be a binary protected attribute, $X \in \mathcal{X}$ to be a set of non-protected features excluding $S$ (i.e., $\mathcal{X} \cap \mathcal{S} = \varnothing$), $Y \in \mathcal{Y}$ to be a binary outcome variable (typically $\mathcal{Y} = \{0, 1\}$), $\hat{Y} \in \mathcal{Y}$ to be an estimator for $Y$ and $Z \in \mathcal{Z}$ to be a proxy for $S$ (in general this means that $Z = f(X)$ for some measurable function $f : \mathcal{X} \mapsto \mathcal{Z}$). $S = s$ means that the variable $S$ takes the value $s$, and similarly for $X$ and $Y$. We also use the shorthand notation $\mathbb{P}(s)$ instead of $\mathbb{P}(S = s)$, and similarly for $Y, \hat{Y}, Z$. We specialize to the case of binary protected attributes S for the sake of simplicity, but our results are more broadly applicable after minor modifications.

## 2 BEST- AND WORST-CASE BOUNDS

Let us now introduce some additional notation: $p_{\hat{y},z,s} = \mathbb{P}(\hat{Y} = \hat{y}, Z = z, S = s)$, $p_{\hat{y},z} = \mathbb{P}(\hat{Y} = \hat{y}, Z = z)$ and $p_{z,s} = \mathbb{P}(Z = z, S = s)$, for all $\hat{y} \in \mathcal{Y}, z \in \mathcal{Z}, s \in \mathcal{S}$. The issue at hand is that of not knowing the joint distribution of $(\hat{Y}, Z, S)$ but simply two bivariate distributions $(\hat{Y}, Z)$ (coming from the model we have built) and $(Z, S)$ (coming from the proxy methodology). Note that the knowledge of these bivariate distributions gives us access to the marginal distributions of $\hat{Y}$, $Z$ and $S$. Furthermore, if $p_{\hat{y},z,s}$ is known then $p_{\hat{y},s} = \sum_{z \in \mathcal{Z}} p_{\hat{y},z,s}$ is available.

**Using linear programming** We thus wish to determine $p_{\hat{y},z,s}$ armed with the knowledge of $\sum_{s \in \mathcal{S}} p_{\hat{y},z,s} = p_{\hat{y},z}$ and $\sum_{\hat{y} \in \mathcal{Y}} p_{\hat{y},z,s} = p_{z,s}$ only. Since these constraints are linear, this can be recast as a linear programming problem.

**Worst- and best-case disparity** Another way of looking at the problem of finding $\mathbf{p} = \{p_{\hat{y},z,s}\}$ is to look for bounds on disparity between protected categories. Recall that disparity is defined as $\sum_{\hat{y} \in \mathcal{Y}} \hat{y} p_{\hat{y}|s} - \sum_{\hat{y} \in \mathcal{Y}} \hat{y} p_{\hat{y}|s'}$ for $s, s' \in \mathcal{S}$ and $s \neq s'$. But this can be rewritten as

$$\text{Disparity}(s, s') = \sum_{\hat{y} \in \mathcal{Y}, z \in \mathcal{Z}} \frac{\hat{y}}{p_s} \cdot p_{\hat{y},z,s} - \sum_{\hat{y} \in \mathcal{Y}, z \in \mathcal{Z}} \frac{\hat{y}}{p'_s} \cdot p_{\hat{y},z,s'}. \tag{1}$$

Given that these are linear combinations of elements of $\mathbf{p}$, applying the same vectorization, we thus have two vectors in $\mathbb{R}^{|\mathcal{Y}| \times |\mathcal{Z}| \times |\mathcal{S}|}$, $\mathbf{d}_s$ and $\mathbf{d}_{s'}$, that encode this relationship:

$$\text{Disparity}(s, s') = \mathbf{d}_s^T \mathbf{p} - \mathbf{d}_{s'}^T \mathbf{p} = (\mathbf{d}_s - \mathbf{d}_{s'})^T \mathbf{p}. \tag{2}$$

We now wish to find upper and lower bounds on the actual disparity given the observations linked to the proxy

$$\max_{\mathbf{p} \geq 0} \left( \mathbf{d}_s - \mathbf{d}_{s'} \right)^T \mathbf{p}$$
$$\text{s.t. } \mathbf{Cp} = \mathbf{s}$$

*Remark* 1. Here $\mathbf{C}$ and $\mathbf{s}$ encodes equality constraints on marginal distributions $p_{\widehat{y},s}$ and $p_{z,s}$ that are known. For more details please refer to section A.4 of Appendix. These are traditional linear programming problems that can be solved via the simplex algorithm. In other words, we have managed to find the interval within which the true disparity lies.

## 3 THE MAXENT SOLUTION

A possible avenue is to add a criterion to optimize, which can be chosen to be the entropy (in other words, minus the KL divergence with respect to a uniform distribution). This is thus a MaxEnt problem (entropy is itself concave):

$$\max_{\mathbf{p} \geq 0} \text{Ent}(\mathbf{p})$$
$$\text{s.t. } \mathbf{Cp} = \mathbf{s}$$

Maximum entropy solutions have appreciable properties and have been thoroughly investigated in finance (see (Avellaneda, 2009) for an overview); it corresponds to the principle of maximum entropy according to which the probability distribution best describing the current state of knowledge is the one with highest entropy.

## 4 RESULTS

We consider the "Adult Income" and "Default of credit card clients" datasets from Dua & Graff (2017). In both datasets, we consider the protected attribute, $S$, of sex, and its corrupted version, $Z$, which is given as

$$Z = \begin{cases} S, & \text{with } p \\ S', & \text{with } 1 - p, \end{cases}$$

for some probability $p$, Here, $S$ is the original value, and $S'$ is the alternative value. We use a logistic regression model to make predictions $\hat{Y}$, based on $Z$. In Figure 1 we plot the signed reconstruction error for $p_{\widehat{y},s}$ that one achieves when varying the corruption amount for both MaxEnt and Worst-/Best- solutions. Besides reconstruction error, we have also looked at how demographic parity (Calders & Verwer, 2010) evolves when changing the corruption amount applied to $S$. The results for this can be found in Section A.7 of Appendix.

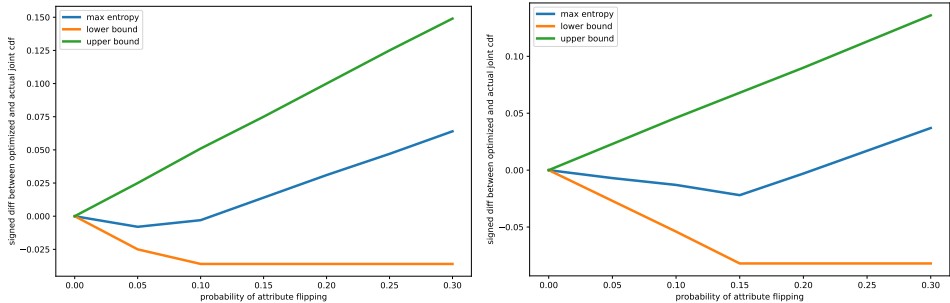

Figure 1: The signed reconstruction error as the function of the flipping probability applied to protected attribute $S$. Left (Adult Income) and Right (Credit Default).

## 5 CONCLUSION

Using linear programming and entropy regularisation, we have showed how to derive best- and worst-case bounds on fairness metrics when protected attributes are only partially observed. The connection to optimal transport in higher dimensions is of interest for further research.

## 6 URM STATEMENT

Author Peter Hill meet the URM criteria of ICLR 2023 Tiny Papers Track.

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

## A  Appendix

### A.1  Background

The absence of discrimination, both direct and indirect, in financial applications, amongst which lending is perhaps the most obvious one, is now a well established principle, as is evidenced by regulation in the United States (cf. (US Congress, 1974; 1968) for example) but also in other countries (see (United Kingdom, 2010; 2018) in the United Kingdom).

Machine learning is now being used at scale in finance, in important decision-making processes such as automated decisioning on granting credit products (Chen, 2018; Turner & McBurnett, 2019), increasing the requirements for fairness-aware algorithms and models (refer to (Information Commissioner's Office, 2020) for a practitioner-oriented overview of challenges attached to developing fair machine learning). Much energy is spent on benchmarking existing approaches ((Agrawal et al., 2020)) and developing new tools ((Turner & McBurnett, 2019; Kim et al., 2020)).

In practice, however, the situation can be rather complex (see (Information Commissioner's Office, 2020; Kurshan et al., 2020) for a description of challenges and unavoidable complications, ranging from accuracy-fairness trade-offs to model governance for machine learning pipelines), and we thus wish to discuss here some difficulties related to uncertainty and ambiguity in developing a fairness framework.

### A.2  Fairness metrics

Many definitions of fairness exist and have been carefully reviewed in other articles (Narayanan, 2018; Verma & Rubin, 2018; Berk et al., 2018; Kim et al., 2020). We will limit ourselves to only a few of these in this paper, such as demographic parity. The concept of demographic parity can be defined more broadly as requesting $\mathbb{P}(\widehat{y}|s) = \mathbb{P}(\widehat{y})$ for all $\widehat{y} \in \mathcal{Y}, s \in \mathcal{S}$, which is the same as the independence of $\hat{Y}$ and $S$. Similarly, equalized odds demands the independence of $\hat{Y}$ and $S$ *conditional* on $Y$, i.e., $\mathbb{P}(\widehat{y}|s, y) = \mathbb{P}(\widehat{y}|y)$ for all $\widehat{y}, y \in \mathcal{Y}, s \in \mathcal{S}$.

### A.3  Correction to conditional independence

First, we derive directly an expression for the joint distribution of $\hat{Y}$ and $S$ as a function of observations involving the proxy variable $Z$. Note that this result is implicit in (Chen et al., 2019).

**Proposition 1.** *For all $\widehat{y} \in \mathcal{Y}, s \in \mathcal{S}$, the following equality holds:*

$$\mathbb{P}(\hat{Y} = \widehat{y}, S = s) = \mathbb{E}\left[\mathbb{P}(\hat{Y} = \widehat{y}|Z)\mathbb{P}(S = s|Z)\right] + \mathbb{E}[cov(\boldsymbol{I}_{\hat{Y}=\widehat{y}}\boldsymbol{I}_{S=s}|Z)].$$

The covariance term adjustment is due to the fact that $\hat{Y}$ and $S$ are in general not independent conditionally on $Z$. This implies that some traditional estimators making that underlying assumptions (cf. weighted estimators described in (Chen et al., 2019)) may be biased. This is related to the concept of exclusion restriction for instrumental variables in the econometric literature.

### A.4  Linear Programming

One can vectorise the tensor $\mathbf{T} = [p_{\widehat{y},z,s}]_{\widehat{y}\in\mathcal{Y}, z\in\mathcal{Z}, s\in\mathcal{S}}$ into a vector $\mathbf{p}$ of dimension $|\mathcal{Y}| \times |\mathcal{Z}| \times |\mathcal{S}|$ (i.e., $\mathbf{p} = \mathbf{vec}(\mathbf{T})$). Similarly, the matrix $[p_{\widehat{y},z}]_{\widehat{y}\in\mathcal{Y}, z\in\mathcal{Z}}$ can be vectorised into a vector $\mathbf{q}$ of dimension $|\mathcal{Y}| \times |\mathcal{Z}|$, and the matrix $[p_{z,s}]_{z\in\mathcal{Z}, s\in\mathcal{S}}$ into a vector $\mathbf{r}$ of dimension $|\mathcal{Z}| \times |\mathcal{S}|$. Lastly, the linear relationships describing the bivariate distributions $\sum_{s\in\mathcal{S}} p_{\widehat{y},z,s} = p_{\widehat{y},z}$ can be encoded into a matrix $\mathbf{A} \in \mathbb{R}^{(|\mathcal{Y}|\times|\mathcal{Z}|)\times(|\mathcal{Y}|\times|\mathcal{Z}|\times|\mathcal{S}|)}$, such that $\mathbf{Ap} = \mathbf{q}$; similarly, we encode the relationship $\sum_{\widehat{y}\in\mathcal{Y}} p_{\widehat{y},z,s} = p_{z,s}$ via a matrix $\mathbf{B} \in \mathbb{R}^{(|\mathcal{Z}|\times|\mathcal{S}|)\times(|\mathcal{Y}|\times|\mathcal{Z}|\times|\mathcal{S}|)}$ such that $\mathbf{Bp} = \mathbf{r}$. Again, this is nothing more than vectorizing the available data.

We can now stack the matrix $\mathbf{A}$ on top of $\mathbf{B}$ to create a matrix $\mathbf{C} \in \mathbb{R}^{(|\mathcal{Y}|\times|\mathcal{Z}|+|\mathcal{Z}|\times|\mathcal{S}|)\times(|\mathcal{Y}|\times|\mathcal{Z}|\times|\mathcal{S}|)}$ and stack the vector $\mathbf{q}$ on top of $\mathbf{r}$ to create a vector $\mathbf{s} \in \mathbb{R}^{|\mathcal{Y}|\times|\mathcal{Z}|+|\mathcal{Z}|\times|\mathcal{S}|}$, such that $\mathbf{Cp} = \mathbf{s}$.

To summarise, we are looking for a vector $\mathbf{p} \geq 0$ such that $\mathbf{Cp} = \mathbf{s}$. Notice that without further information, the vector $\mathbf{p}' = \mathbf{vec}\left(\left[\frac{p_{\hat{y},z} \cdot p_{z,s}}{p_z}\right]_{\hat{y} \in \mathcal{Y}, z \in \mathcal{Z}, s \in \mathcal{S}}\right)$, i.e., the vector made up of entries where have supposed that $\hat{Y}$ and $S$ are independent conditionally on $Z$, is a solution.

### A.5 BEST- AND WORST-CASE BOUNDS

The main text only indicates the worst-case bound, but we indicate here how to derive the best-case bound too.

$$\text{Disparity}(s, s') = \mathbf{d}_s^T \mathbf{p} - \mathbf{d}_{s'}^T \mathbf{p} = (\mathbf{d}_s - \mathbf{d}_{s'})^T \mathbf{p}. \tag{3}$$

We now wish to find upper and lower bounds on the actual disparity given the observations linked to the proxy

$$\max_{\mathbf{p} \geq 0} (\mathbf{d}_s - \mathbf{d}_{s'})^T \mathbf{p}$$
$$\text{s.t. } \mathbf{Cp} = \mathbf{s}$$

and

$$\max_{\mathbf{p} \geq 0} (\mathbf{d}_{s'} - \mathbf{d}_s)^T \mathbf{p}$$
$$\text{s.t. } \mathbf{Cp} = \mathbf{s}.$$

*Remark* 2. Note that this approach can tackle any concave function $f$ of the probability vector $\mathbf{p}$. It is also worth pointing out that our proposed approach is closely related to optimal transport in higher dimensions.

### A.6 REGULARISED MAXENT

One can add some regularisation to the MaxEnt problem by adding a divergence constraint on $\mathbf{p}$ to ensure that $\mathbf{p}$ is not too "wild". This can be achieved by adding a KL term between $\mathbf{p}$ and $\mathbf{p}'$, leading to

$$\max_{\mathbf{p} \geq 0} (\mathbf{d}_s - \mathbf{d}_{s'})^T \mathbf{p}$$
$$\text{s.t. } \mathbf{Cp} = \mathbf{s}, \text{KL}(\mathbf{p}, \mathbf{p}') \leq \varepsilon,$$

with $\varepsilon > 0$ a tolerance level. This has a Bayesian interpretation whereby our prior on $\mathbf{p}$ is $\mathbf{p}'$.

### A.7 ADDITIONAL RESULTS

Here we show how demographic parity changes as different amount of corruption applied to the protected attribute $S$. As one can see there is a point where demographic parity becomes zero, suggesting that we have achieved full randomization.

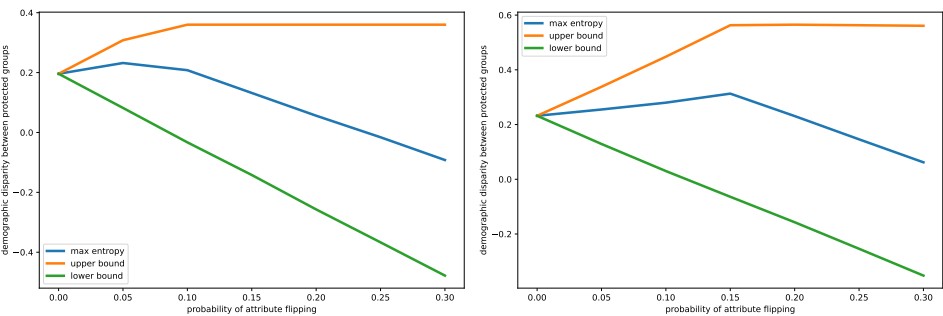

Figure 2: Demograpic parity as the function of flipping probability applied to protected attribute $S$. Left (Adult) and Right (Credit Default).

