# OpenReview forum: "Fairness Under Partial Observability"
_ICLR.cc/2023/TinyPapers — Submitted to Tiny Papers @ ICLR 2023_

### Official Review · Reviewer_twQV · 2023-03-26

**Confidence:** 3

**Summary Of Contributions:**

The authors discuss the challenges of implementing group privacy for ML models when certain aspects of the system and model are unknown or uncertain due to protection of sensitive features.

**Rating:**

High Potential (HP): a submission which meets the reviewing criteria and has potential to make an impact on the field

**Strengths And Weaknesses:**

Strengths:

1. The authors consider an interesting problem that identifies how protected features (such as sex or race) can impact the fairness of a model. In particular, the paper attempts to identify how the protection mechanism affects the probabilistic fairness disparity (Eqns. (1) and (2)).

2. If the true feature is $s$, the protected version of $s$ is $z$ and the predicted model response is $\hat{y}$ then, the authors wish to identify the joint distribution $p_{\hat{y}, z, s}$; given only the joint distributions $p_{\hat{y}, z}, p_{s, z}$. By computing this distribution we could compute the disparity as well as the demographic parity.

3. The authors show that such fairness measures can be easily bounded (not tight) from both ends and it's possible to find the probability for which we get the best demographic parity (best fairness).

Weaknesses:

1. The paper's motivation is only partially clear in this paper. Some portions such as the impact on privacy by the suggested mechanism are unclear.

2. Further experimentation is necessary to evaluate the limitations of the mechanism and also find ways to improve the bounds. Certain design choices like entropy or the use of uniform distribution as a baseline could be described further.

3. Current approach is limited by probabilistic flipping. Other differential privacy mechanisms like the exponential mechanism should be considered. Also, the authors can clarify what other kinds of protection mechanisms they consider for this paper.


**Suggested Changes:**

1. The authors should consider adding further motivation for why this problem is essential. Although knowing a good flipping probability can help improve the fairness measure, it requires multiple experiments, controlling the flipping probability, and might lead to further data leakage (by accessing the data multiple times for the experiments). Furthermore, it might not be possible to control the flipping probability for privacy mechanisms like differential privacy as this will also affect the privacy guarantee. Thus, a joint privacy and fairness measure needs to be considered.

2. The experimental results are promising but they should be expanded to include multi-class ($> 2$) features.

---

### Official Review · Reviewer_8sFB · 2023-03-29

**Confidence:** 5

**Summary Of Contributions:**

The authors address the challenge of implementing group fairness-aware models when protected attributes are only partially observable. They use linear programming and entropy regularization to derive best- and worst-case bounds on fairness metrics.

**Rating:**

High Potential (HP): a submission which meets the reviewing criteria and has potential to make an impact on the field

**Strengths And Weaknesses:**

The authors address an important and intriguing real-world problem: how to achieve model fairness when dealing with partially observed attributes. The paper is well-structured and effectively communicates its key points. The methodology's notation and derivation are presented in a clear and easily understandable manner. Here is a summary of the strengths and weaknesses of the paper.

Strengths:

1. The paper tackles a realistic and substantial problem setting.
2. The methodology's derivation is thorough, logically sound, and accessible to readers.
3. The results of the method are persuasive and impactful.
4. The method can be generalized with minimal modification.

Weaknesses:

1. Several essential details necessary for understanding the method are located in the appendix, which might cause some confusion when reading the main paper.
2. The paper discusses the best and worst bounds on the disparity of protected features; however, only the worst-case scenario is presented in the main text. Including both best and worst-case scenarios in the main text could enhance the reader's understanding.
3. The paper lacks a comprehensive literature review, which could be improved by discussing how other researchers have approached the same or similar problems.

**Suggested Changes:**

1. In Section 2, it would be clearer for readers if the authors referenced that the best-case scenario is discussed in the appendix.
2. The paper would be more comprehensive if the authors included a discussion of related works.

---

### Official Review · Reviewer_yv3L · 2023-04-02

**Confidence:** 4

**Summary Of Contributions:**

This paper focuses on the fairness under the partial observability of protected attributes and provides the theoretical analysis for fairness under partial observability.

**Rating:**

High Potential (HP): a submission which meets the reviewing criteria and has potential to make an impact on the field

**Strengths And Weaknesses:**

**Strengths**
1. This paper is well-written and easy to follow and comprehend the proposed ideas and methodologies.
2. The fairness under partial observability is super important in real applications, as it reflects the real-world limitations often encountered when dealing with sensitive or incomplete data.
3. The theoretical analysis in the paper seems sound to me.

**Weaknesses** \
I don’t see obvious weaknesses in this paper.


**Suggested Changes:**

N/A

---

### Author Response · Authors · 2023-06-01
**Opt-in for achival**

The authors wish to opt-in for archival in the ICLR Tiny Papers Track.

---

### Meta-Review · Area_Chair_gjVi · 2023-04-08

**Recommendation:** Invite to present
**Confidence:** 4

**Metareview:**

This paper discusses a challenge for group fairness-aware models when information about protected attributes are not fully observable, due to the introduction of a proxy. Given information about the accuracy of the proxy, and about the predictions using the proxy, the authors show how to infer possible relationships between the true protected attribute and the predictions using linear programming, focusing on best- and worst-case bounds on privacy metrics as well as a maximum-entropy criterion.

All of the reviewers found the paper to be well-written and technically sound, and additionally described the problem setting as realistic and practically applicable. Reviewer 8sFB also notes that it could be easily generalized to new settings.

On the other hand, the reviewers identified a few limitations of the work, in particular about the overall framing and structure.
Reviewer 8sFB points out that important details about the method are only described in the appendix and are not obvious from the main paper, and also notes that the paper would benefit from a more thorough literature review and a discussion of the worst-case bound in the main paper.
Reviewer twQV found the motivation for the proposed approach to be partially unclear. They also note that the interaction between the fairness metrics and the chosen proxy could be discussed more, and that additional experiments would be useful.

Overall the reviewers agree that this work meets the Clear, Correct, and Reproducible criteria and has high potential, so I recommend that it be invited for presentation.

**Summary:**

The paper discuses how to infer fairness metrics in a partially observable setting using linear programming. Reviewers all agree it meets the CCR criteria and has high potential.

**Comments And Feedback To The Authors:**

One thing that was a bit unclear to me on reading the submission: are the upper and lower bounds on disparity primarily intended to be used for directly evaluating fairness of a method, e.g. by looking for a method with a good set of bounds? Or, are those bounds just intended as heuristics for inferring hypothetical joint probability distributions $p_{\hat{y},s}$, which can then be applied for some other purpose?

**Reason For Not Giving A Higher Recommendation:**

Reviewers 8sFB and twQV have noted a few ways that the presentation of the results could be improved: the authors could give more motivation for the proposed bounds and choice of corrupted proxy Z, and also could move some of the essential details from the appendix to the main paper. I also found it somewhat confusing that the upper and lower bounds were presented in terms of disparity, but that the results section focused on evaluating reconstruction error, not disparity.

**Reason For Not Giving A Lower Recommendation:**

All reviewers agree that the method meets the "Clear, Correct, and Reproducible" criteria for the tiny papers track.

---

### Decision · Program_Chairs · 2023-04-09

Invite to present